# Asking What Matters: Reward-Driven Clarification for Software Engineering Tasks

**Sanidhya Vijayvargiya** [1]  **Vijay Viswanathan** [1]  **Graham Neubig** [1]

## Abstract

Humans often specify tasks incompletely, so assistants must know when and how to ask clarifying questions. However, effective clarification remains challenging in software engineering tasks as not all missing information is equally valuable, and questions must target information users can realistically provide. We study clarification in real software engineering tasks by quantifying which types of information most affect task success and which questions elicit useful responses from simulated users. Using Shapley attribution and distributional comparisons, we identify two key properties of effective clarification: task relevance (which information predicts success) and user answerability (what users can realistically provide). We operationalize these properties as multi-stage reinforcement learning rewards to train CLARITI, an 8B-parameter clarification module, that matches GPT-5's resolution rate on underspecified issues while generating 41% fewer questions. Our results suggest that grounding reward design in empirical analysis of information impact and user answerability improves clarification efficiency.

## 1. Introduction

Real-world user requests are frequently underspecified. Despite rapid improvements, LLM-based agents remain brittle when user instructions are underspecified or ambiguous, leading to wasted computation or task failure. Recent empirical studies highlight ambiguity and missing information as dominant failure modes for agents (Zhang et al., 2024; Vijayvargiya et al., 2025), underscoring the need for systems that can detect uncertainty and proactively seek additional information.

A growing body of work demonstrates that clarifica-

---

[1]Language Technologies Institute, Carnegie Mellon University, Pittsburgh, USA. Correspondence to: Sanidhya Vijayvargiya <sanidhyv@cs.cmu.edu>.

*Proceedings of the 43rd International Conference on Machine Learning*, Seoul, South Korea. PMLR 306, 2026. Copyright 2026 by the author(s).

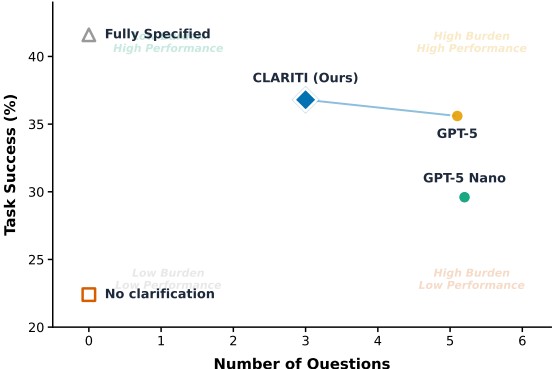

*Figure 1.* Our trained clarification model, CLARITI, achieves GPT-5-level performance (36.80%) with 41% fewer average questions (3.0 vs 5.1) by prioritizing task relevance and user answerability, demonstrating effective clarification at low user burden.

tion questions can substantially improve downstream outcomes (Zhang & Choi, 2025; Chen et al., 2025), motivating work on user modeling that personalizes questions to individuals (Sun et al., 2025), or infers mental states (Zhou et al., 2025). We take a complementary, model-centric perspective: given typical user knowledge boundaries, how should models optimize clarification to maximize task success?

Prior work provides limited understanding of which missing information is most valuable or how agents should prioritize clarification under interaction constraints. The analyses primarily target linguistic ambiguity—often involving a single point of intent-level uncertainty (e.g., resolving a referent or disambiguating an intent)—which does not capture the broader underspecification found in task-oriented settings where multiple types of information may be missing with varying importance (Madge et al., 2025). As a result, despite evidence that clarification helps, we lack a principled understanding of what information to ask about, whether users can answer the questions posed, and how these factors impact task success.

We study clarification in software engineering (SWE), a domain where tasks require precise specifications and objective success can be measured via test suites. SWE issues often omit critical details such as environment configuration, expected behavior, or implementation constraints. An effective agent must prioritize among many possible information needs and identify which gaps genuinely block progress. We

consider a controlled, single-turn setting where the agent poses clarifying questions to a simulated user before beginning implementation. Using separate modules for clarification generation and task execution, we isolate clarification quality from agent problem-solving capabilities allowing us to evaluate clarification independent of code-generation ability. This design exposes two fundamental dimensions: effective clarifications must target information likely to improve task success (*task relevance*) while remaining *answerable* by typical users given their knowledge about the issues they report. We demonstrate that the identified qualities for effective clarification are learnable and predictive of downstream success by training a clarification module, CLARITI (CLARIfication with TIered rewards)[1], that improves downstream agent performance (Figure 1).

We first characterize the information needs common to underspecified SWE tasks, then investigate three research questions. Concretely, our contributions are:

- **RQ1: What information most impacts task success?** We quantify the association between information category availability and downstream agent performance through Shapley value analysis across 700 underspecified SWE instances. Error information exhibits the strongest association, followed by implementation details and environment configurations, revealing a hierarchy where concrete diagnostic information contributes more to agent performance than abstract goal specifications.

- **RQ2: What makes questions answerable?** We investigate characteristics that distinguish answerable from unanswerable clarification questions through distributional analysis of the two classes. We identify that answerable questions tend to ground in observable behaviors, maintain appropriate technical depth, and avoid requesting internal state users would not know. Analyzing clarification composition, we find that performance often plateaus or declines with more questions while imposing higher user burden. Excess questions can waste user effort without improving outcomes, and any informational gains from additional questions are offset by the introduction of irrelevant or unusable information into the context.

- **RQ3: Training with empirically-grounded rewards.** We design a multi-stage reward pipeline targeting both task relevance (from RQ1's impact hierarchy) and answerability (from RQ2's distributional analysis), plus auxiliary criteria of non-redundancy and diversity. Our trained 8B-parameter module matches GPT-5's performance on our task (88% of fully specified upper bound) while generating 41% fewer questions, demonstrating that effective clarification can be learned through principled, empirically-grounded reward design.

---

[1]Code and data can be accessed at https://github.com/sani903/Teaching-Effective-Clarification

Our study connects clarification quality directly to empirical task-solving outcomes, providing a methodology for identifying what information matters, understanding what makes questions answerable, and training models to optimize both dimensions. While our work focuses on software engineering, the framework combines empirical impact analysis, answerability characterization, and reward-driven training that can generalize to domains where agents rely on user-provided specifications.

## 2. Information Categorization

Effective clarification requires understanding what information agents lack. While prior work studies linguistic ambiguity (e.g., referential or syntactic ambiguity), it does not categorize the types of missing information that affect agent-based task resolution. We address this gap by analyzing underspecified issues in SWE-Bench. We define an information need as information absent from the issue description that is required for an agent to produce a correct patch without relying on unverifiable assumptions.

### 2.1. Categories of Information Needs

We derive our categorization from naturally occurring underspecification in SWE tasks. We begin with the expert annotations of SWE-Bench issues which were used to construct the Verified subset (Chowdhury et al., 2024). The experts score each issue's underspecification level (0–3 scale), with higher scores corresponding to greater underspecification. We extract 112 issues with high underspecification (score $\geq 2$), where annotators provide detailed justifications for their ratings. These justifications describe specific information gaps that prevent resolution, such as missing error messages, unclear expected outputs, or absent reproduction steps. Issues often contain multiple information gaps. The codebook was developed by two authors through an iterative process. The authors independently coded the first 30 issues and then compared category assignments, resolving minor differences through discussion. This was followed by annotation of the remaining issues in batches following qualitative research principles (MacQueen et al., 1998). The resulting categorization comprises six categories of information needs (Table 1). Each category represents a distinct type of missing information that agents may need to obtain through clarification. Expert involvement is limited to this one-time codebook development over a limited number of samples and per-instance category annotation can be fully automated via LLM judges, making the framework scalable.

## 3. Experimental Setup

We describe the datasets, agent framework, and evaluation methodology used throughout this work. All subsequent experiments (RQ1–RQ3) build on this shared infrastructure.

*Table 1.* We identify six categories of information needs from 112 underspecified SWE-Bench Verified annotations, ranging in frequency from occurring in as many as 65% of issues to only 3% of issues.

| Category | Frequency | Core Question | Example Missing Information |
|---|---|---|---|
| **Error Information** | 65% | What specific failure is occurring? | Missing stack trace showing where code crashes; absent error message revealing constraint violation; unclear description of incorrect output format. |
| **Expected Behavior** | 33% | What behavior should occur instead? | Missing specification of correct return value; absent description of intended output format; unstated desired system state after operation completes. |
| **Implementation Details** | 37% | What approach or constraints should guide the implementation? | Missing step-by-step guidance on achieving expected behavior; lack of exact functions to modify and how they should be modified; unclear whether to change existing logic or implement from scratch. |
| **External References** | 15% | What external context is needed to understand requirements? | Inaccessible API documentation link; missing reference to upstream library behavior; absent dataset schema or format specification. |
| **Reproduction Steps** | 12% | Under what conditions does the issue manifest? | Missing command-line invocation that triggers the bug; absent minimal code example demonstrating the failure; unclear input that produces incorrect behavior. |
| **Version/Environment** | 3% | What configuration details affect the issue? | Missing dependency version that exhibits the bug; absent OS or Python version; unclear configuration flags or environment variables. |

## 3.1. Categorization-Grounded Evaluation Dataset

We construct controlled underspecification variants of SWE-Bench Verified issues to enable systematic measurement of information impact. Starting from 500 issues, we first annotate which information categories appear in each issue description using the codebook introduced in Section 2.1. Annotation is performed with GPT-5 (Singh et al., 2025) using structured prompts that apply the codebook definitions (Appendix A.2). For each issue, we then generate three underspecified rewrites by removing subsets of the annotated categories. Specifically, we randomly select subsets of categories present in the original issue and instruct GPT-5 to produce natural-language rewrites that omit those categories while preserving the remaining information. Each issue therefore yields three distinct underspecified variants reflecting different omission patterns. We manually validate 50 randomly sampled rewrites to verify that they (1) omit the intended categories, (2) preserve information from non-target categories, and (3) remain plausible GitHub issue descriptions (subsection A.3). The full generation process yields 1,500 underspecified issue variants (500 issues × 3 rewrites). For impact analysis (RQ1), we evaluate a random sample of 700 instances from this pool, ensuring coverage across different combinations of missing information. For RQ2 and RQ3, we select one rewrite per SWE-Bench Verified issue to construct a dataset of 500 underspecified issues.

## 3.2. Agent Framework and Evaluation Protocol

**Agent environment.** We use the OpenHands framework (Wang et al., 2024) with Seed OSS 36B In-struct (ByteDance Seed Team, 2025) as the agent backbone. We select Seed OSS 36B because it represents a strong open-weight coding model at the time of our experiments, enabling reproducible evaluation without reliance on proprietary APIs. OpenHands provides a sandboxed environment in which agents can edit files, execute bash and Python commands, and iteratively refine solutions. Agents are configured with a maximum of 30 interaction iterations and a decoding temperature of 0.3. The agent backbone and configuration remain fixed across all experiments to isolate the impact of clarification strategies.

**Evaluation protocol.** For each evaluation instance, we measure binary task success based on whether the patch produced by the agent passes all repository test cases after application. This metric provides an objective measure of downstream task completion. In RQ1, agents receive the underspecified issue descriptions directly without clarification to measure the baseline impact of missing information. In RQ2 and RQ3, we introduce clarification modules that allow the agent to ask questions before beginning implementation. Questions are answered by a simulated user implemented with GPT-5, which has access to the fully specified version of the issue similar to Vijayvargiya et al. (2025). The answers are then provided to the agent as additional context prior to code generation.

## 4. RQ1: Impact of Information Categories on Task Success

Having categorized the types of missing information, we next examine how these categories relate to agent task suc-

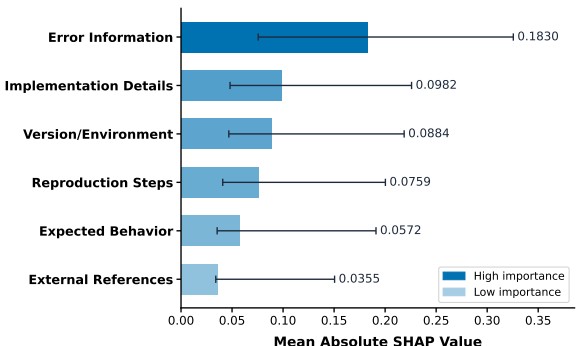

*Figure 2.* Mean absolute SHAP values with 95% bootstrap confidence intervals measuring the association between each information category and task success.

cess. Specifically, which categories of missing information are most associated with successful task resolution?

To estimate the predictive contribution of each category, we perform Shapley value analysis (Lundberg & Lee, 2017). For each of the 700 evaluation instances, we represent the available information as a binary feature vector indicating whether each category is present (i.e., not hidden). We train predictive models that map these feature vectors to binary task success and compute SHAP values to estimate the marginal contribution of each category while accounting for interactions among features (details in Appendix A.4). We report bootstrap confidence intervals computed from 10,000 resamples to characterize uncertainty in the estimates.

Before analyzing individual categories, we first measure the overall effect of underspecification. Converting issues from fully specified to underspecified variants reduces agent performance from 43.8% to 23.7% success across the 700 evaluated instances, confirming that missing information substantially reduces task success. Figure 2 reports the SHAP attribution results. *Error Information* has the highest mean SHAP value (0.183), followed by *Implementation Details* (0.0982). *Expected Behavior* (0.0572) and *External References* (0.0355) show lower mean contributions. Bootstrap confidence intervals overlap across categories, reflecting the limited sample size; we therefore interpret the ordering as an indicative trend rather than a statistically significant ranking. Despite this uncertainty, the relative ordering differs from the frequency with which categories are missing in naturally occurring underspecified issues (Table 1). For example, *Expected Behavior* is the most frequently missing category (65% of issues) yet shows only moderate association with task success. Conversely, *Error Information* is absent in 33% of underspecified issues but has the highest mean SHAP value. One possible explanation is that concrete failure signals such as stack traces or error messages provide localized diagnostic information (e.g., file paths or failure points) that may help agents narrow the search space when debugging. *Implementation Details* follows a similar pattern: although missing in 37% of underspecified issues,

it has relatively high predictive contribution, suggesting that explicit guidance on how a fix should be implemented may reduce ambiguity during solution search. An interesting case is *Version/Environment*, which is missing in only 3% of issues yet shows a comparatively high mean SHAP value, indicating that when such details are absent they may disproportionately affect task outcomes.

Overall, these results suggest that the information most frequently omitted in issue descriptions does not necessarily align with the information most strongly associated with agent success. Categories providing concrete diagnostic signals or implementation guidance tend to show higher predictive contribution than categories describing expected outcomes or external context. This impact hierarchy motivates the impact-weighted reward signals used to train our clarification model in RQ3.

## 5. RQ2: User-Answerable Clarification Questions

RQ1 identified which categories of missing information most strongly influence task success. We now examine a complementary dimension of clarification quality: *answerability*. Even when a question targets high-impact information, it provides little practical value if users cannot supply the requested information. This introduces a practical tension in question formulation. Highly generic questions (e.g., *Can you provide more details?*) are easy for users to answer but provide limited guidance, while overly technical questions (e.g., *What is the internal state of the cache manager?*) may target useful information that typical users cannot reasonably provide.

Rather than modeling individual user expertise, we analyze structural characteristics that distinguish questions that users can plausibly answer from those that request information beyond typical user knowledge. Specifically, we study distributional differences between answerable and unanswerable clarification questions and investigate how the composition of clarification sets affects downstream task success.

### 5.1. Experimental Setup

We use the same 500 underspecified issues from the taxonomy-grounded dataset introduced in RQ1. For each issue, we generate clarification questions using two models representing different parameter scales and code understanding capabilities: GPT-5 and GPT-5 nano (Singh et al., 2025). Both models are prompted to produce clarification questions targeting missing information in the underspecified issue (full prompts provided in Appendix A.5).

To assess answerability, we compare each generated question against both the underspecified issue and its corresponding fully-specified original issue. We operationalize answer-

*Table 2.* Four themes derived from distributional characteristics distinguishing answerable from unanswerable clarification questions.

| Strategy | Description | Answerable Example | Unanswerable Example | $V'$ | N |
|----------|-------------|--------------------|--------------------|------|---|
| **Ground in Evidence** | Request concrete artifacts users can directly share | *Share the stack trace* | *What is the middleware execution order?* | 0.111 | 23 |
| **Demand Specificity** | Ask for precise values rather than abstract descriptions | *Which Python version?* | *What is the optimal version?* | 0.081 | 15 |
| **Minimize Scope** | Request the smallest information unit isolating the issue | *Provide a 10-line script demonstrating this* | *Describe your entire architecture* | 0.072 | 13 |
| **Ensure Actionability** | Focus on actions users can perform or directly observe | *Run* `pytest -v` *and share output* | *What would happen if you refactored?* | 0.093 | 9 |

$V'$ = median effect size; N = discoveries per theme (60 total). All $p < 0.05$. Complete results in Appendix A.6.

ability relative to the information contained in the issue documentation. A question is considered answerable if it requests information that is missing from the underspecified issue but present in the fully-specified version.

We use GPT-5 as an automatic judge to perform this classification. The judge receives both the underspecified issue (the context available to the question-generating model) and the fully-specified issue (the reference source of complete information). It then assigns each question to one of three categories:

- **Answerable**: The requested information appears in the full issue but is absent from the underspecified issue (i.e., the question correctly targets missing information).

- **Unanswerable**: The requested information does not appear in the fully-specified issue (i.e., the question requests information that is unlikely to be available to the user).

- **Redundant**: The requested information already appears in the underspecified issue.

Answerability is approximated using the full issue as a proxy. Real user capability may be lower or higher than this proxy. This setup distinguishes questions that correctly request missing information from those that either ask for information already provided or request information that is not available in the issue context. For the analyses in this section, we focus on the distinction between **answerable** and **unanswerable** questions, leaving redundant questions to the analysis in RQ3. In total, we analyze clarification questions for 500 issues per model.

### 5.2. What Makes Questions Answerable?

To understand what differentiates answerable from unanswerable questions, we employ distributional analysis (Zhong et al., 2023) to identify linguistic, structural, and semantic features that differ significantly between the two groups. Across GPT-5 and GPT-5 Nano outputs, we measure distributional differences using the Vargha–Delaney effect size ($V'$), identifying 60 significant characteristics ($p < 0.05$) that group into four strategic themes (Table 2).

These strategies are orthogonal to the information taxonomy from RQ1: while RQ1 identifies *which* categories matter, these themes describe *how* to formulate questions so users can realistically answer them. A poorly formulated question targeting a high-impact category may still be unanswerable (e.g., *What internal error handling logic failed?*), while a well-formulated question on a lower-impact category may not (e.g., *What URL appears in the error message?*). Effective clarification thus requires optimizing both dimensions jointly. Consistent patterns across both models suggest these strategies reflect general properties of answerable question formulation rather than model-specific artifacts (full results in Appendix A.6).

### 5.3. Impact of Answerability on Performance

Beyond individual questions, we also examine how the overall composition of clarification sets influences downstream task success. In particular, we study the relationship between the number of questions asked, the proportion of answerable questions, and task success.

Figure 3 shows that increasing the number of questions does not consistently improve performance. Success rates plateau despite larger clarification sets. At the same time, the proportion of answerable questions decreases as question count increases. This pattern suggests that asking more questions does not guarantee better outcomes, and that the informational value of additional questions may be offset by the introduction of low-quality or unanswerable ones.

These observations highlight the importance of concentrating clarification effort on a small number of high-quality questions. Clarification sets with a higher proportion of answerable questions appear to correlate with stronger task performance while reducing user burden.

Together with RQ1's information impact hierarchy (which identifies *which* information categories matter most), these findings motivate the training strategy introduced in RQ3. Specifically, we design reward signals that encourage models to target high-impact information while maintaining a

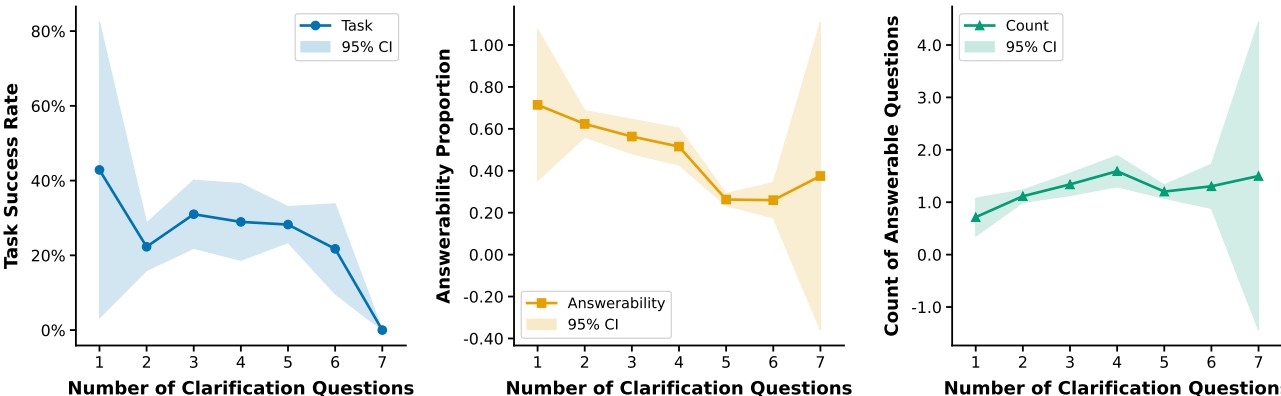

*Figure 3.* Task success as a function of the number of clarification questions asked. Performance plateaus as question count increases, while the proportion of answerable questions declines.

high proportion of answerable questions, thereby prioritizing both informational relevance and user accessibility.

# 6. RQ3: Do Empirically-Grounded Objectives Improve Clarification?

RQ1 identifies which types of missing information most strongly impact task success, while RQ2 characterizes which questions users can realistically answer. We now evaluate whether these principles, when directly utilized as training objectives, are sufficient to induce improved clarification behavior in a learned model. While multiple components contribute to overall performance, we design RQ3 to primarily evaluate the impact of the learning objective. We therefore keep the agent pipeline, dataset, and evaluation setup fixed, and introduce reward signals derived from RQ1 and RQ2 as the primary source of variation.

## 6.1. Training Data and Model

Training data is constructed from SWE-Gym Raw (Pan et al., 2025). Using DeepSeek-V3 (DeepSeek-AI et al., 2025), we generate underspecified variants of real GitHub issues through a four-step pipeline: (i) removing selected information from the original issue, (ii) identifying missing information relative to the full issue, (iii) converting these gaps into clarification questions, and (iv) filtering questions which do not have grounding in the underspecified issue (containing entities only mentioned in the full issue) to prevent hallucinations. This yields 3,000 supervised fine-tuning (SFT) pairs and 1,000 distinct instances for reinforcement learning for training Qwen3 8B model (Yang et al., 2025).

We first perform SFT to establish an initial capability for identifying missing information. Post-SFT, our model exhibits two main failure modes: it frequently asks about details already mentioned in the task description, and it uses generic question templates rather than adapting to issue-specific content. We address both through reinforcement

learning with GRPO (Shao et al., 2024), optimizing directly for the properties identified in RQ1 and RQ2 via rubric-style rewards (Viswanathan et al., 2025; Dineen et al., 2025).

## 6.2. Four-Stage Reward Pipeline

A natural approach is to use downstream task success directly as a training reward. However, this is computationally intractable for clarification: each training step would require running full agent trajectories (up to 30 state-action pairs of thousands of tokens each) per generated clarification set, with no decomposable signal distinguishing why a clarification helped or failed. Our intrinsic reward pipeline is designed to circumvent this bottleneck, using RQ1's Shapley analysis to ground task relevance as a principled proxy for outcome, and RQ2's distributional analysis to capture answerability constraints that outcome-based RL would need to discover implicitly, if at all.

We design a four-stage reward pipeline that decomposes clarification quality into four measurable, intrinsic properties: non-redundancy, diversity, answerability, and task relevance (Figure 4). Each stage applies a rejection threshold; candidates failing a stage receive zero reward and are not evaluated in subsequent stages. This structure prioritizes simpler constraints before refining higher-level behavior and critically prevents each reward component from being gamed in isolation by earlier, easier-to-satisfy objectives.

**Stage 1: Non-redundancy** Questions whose answers are already present in the underspecified issue are penalized:

$$r_{\text{non-redundancy}} = 1 - \frac{\text{redundant questions}}{\text{total questions}}.$$

This stage must precede all others because redundant questions trivially satisfy answerability, and would otherwise inflate Stage 3 scores without any genuine information-seeking. Placing this constraint first ensures that reward signals measure what they are intended to measure.

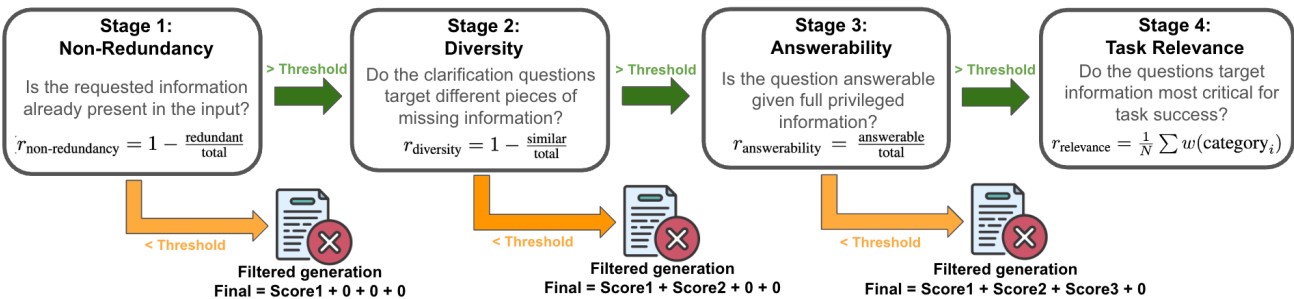

*Figure 4.* Multi-stage reward pipeline. Generated clarification sets flow through four sequential stages: (1) **Non-Redundancy** filters generations with high number of questions answerable from the underspecified issue (threshold $r \geq 0.5$), (2) **Diversity** filters generations with generic questions similar across different issues (threshold $r \geq 0.5$), (3) **Answerability** scores whether users can answer questions in the generation from the full issue, and (4) **Task Relevance** classifies questions in the generation into information categories weighted by empirical impact (RQ1). Final reward equally weights all stages: $r_{\text{final}} = 0.25(r_{\text{redundancy}} + r_{\text{diversity}} + r_{\text{answerability}} + r_{\text{relevance}})$.

**Stage 2: Diversity** Generic questions and multiple questions targeting the same information need are penalized:

$$r_{\text{diversity}} = 1 - \frac{\text{similar questions}}{\text{total questions}}.$$

Stage 1 filtering is insufficient to prevent template collapse: reusable question structures are not redundant with any specific issue, but they reflect a degenerate policy that ignores issue-specific content. Stage 2 penalizes lexical and semantic similarity both within a generation and across cached generations from the same training batch, encouraging the model to move toward questions that reference concrete, issue-specific entities rather than broadly applicable probes.

**Stage 3: Answerability** Questions are scored by whether their answers appear in the fully specified issue:

$$r_{\text{answerability}} = \frac{\text{answerable questions}}{\text{total questions}}.$$

This uses RQ2's finding that effective questions must remain within the knowledge boundaries of typical users. Diversity filtering in Stage 2 helps ensure that the model cannot satisfy answerability by learning question structures that are generically answerable across many issues, rather than by learning to identify what a specific user can provide.

**Stage 4: Task relevance** Questions are classified into information categories (Section 2) and weighted by their empirical importance from RQ1:

$$r_{\text{relevance}} = \frac{1}{N} \sum_{i=1}^{N} w(\text{category}_i).$$

Without this stage, the model distributes questions roughly according to the natural frequency of missing information in the training data, implicitly replicating the misalignment identified in RQ1: the most frequently missing category (Expected Behavior, present in 65% of underspecified issues) has lower predictive contribution to task success than categories missing less often, such as Error Information.

Task relevance weighting helps corrects this by explicitly incentivizing questions that target high-impact information regardless of how commonly it is missing. The final reward equally weights all stages:

$$r_{\text{final}} = \frac{1}{4}(r_{\text{non-redundancy}} + r_{\text{diversity}} + r_{\text{answerability}} + r_{\text{relevance}}).$$

Achieving improvements across all four stages requires striking a balance between the desired properties. Maximizing reward for a particular stage clashes with the objectives for subsequent stages, making simple forms of reward hacking less effective since all stages are equally weighted. The trained model learns to strike trade-offs between the stages to achieve optimal clarification. We assign rewards using Qwen 3 32B as judge across all four stages during training and leverage GPT-5 as judge during evaluation to prevent bias. The tiered reward structure also helps the judge break the complex task of evaluating clarification into simpler sub-tasks which it can complete more reliably.

### 6.3. Downstream Evaluation

We evaluate the trained clarification module within the agent pipeline described in Section 3, keeping all other components fixed. Experiments are conducted on 250 underspecified issues drawn from our categorization-grounded dataset. We compare: (i) no clarification baseline, (ii) GPT-5 Nano, (iii) GPT-5, (iv) our trained model, and (v) a fully specified upper bound. User responses are simulated by extracting answers from the fully specified issue, isolating clarification quality as the primary experimental variable. Importantly, training and evaluation distributions are intentionally separated with training data drawn from SWE-Gym Raw (different repositories from SWE-Bench Verified) and generated with DeepSeek-V3, while all evaluation datasets and judges use GPT-5, preventing overestimation of test set accuracy.

### 6.4. Results

Table 4 reports downstream performance. Without clarification, the agent solves 22.4% of tasks relative to the 41.6%

*Table 3.* Our model achieves 88% of the fully-specified performance ceiling with substantially fewer questions than GPT-5.

| Method | Success↑ | Answerability↑ | Relevance↑ | #Qs↓ |
|---|---|---|---|---|
| No clarif. | 22.4 | — | — | — |
| Nano | 29.6 | .339 | .576 | 5.2 |
| GPT-5 | 35.6 | .369 | .580 | 5.1 |
| **CLARITI** | **36.8** | **.373** | **.622** | **3.0** |
| Full issue | 41.6 | — | — | — |

*Table 4.* Ablation results. Removing either Stage 3 or Stage 4 substantially degrades task success, and SFT alone provides minimal gains over no clarification.

| Method | Success↑ | Answerability↑ | Relevance↑ | #Qs↓ |
|---|---|---|---|---|
| No clarif. | 22.4 | — | — | — |
| SFT | 24.2 | .382 | .583 | 1.5 |
| w/o Stg 3 | 24.4 | .341 | .613 | 2.5 |
| w/o Stg 4 | 32.0 | .390 | .596 | 2.2 |
| All Stg | **36.8** | **.373** | **.622** | **3.0** |

fully specified baseline, confirming that missing information substantially impairs completion. All clarification methods improve over the baseline, but differ substantially in question efficiency.

Our model achieves 36.8% task success with only 3.0 questions on average—41% fewer than GPT-5 (5.1)—recovering 88% of fully specified performance. Despite similar answerability scores (0.373 vs. 0.369), the key differentiator is category allocation: our model directs 26.4% of questions toward Error Information (the highest-impact category) compared to 10.2% for GPT-5 and 6.0% for GPT-5 Nano, while reducing lower-impact categories such as Reproduction Steps (30.2% → 19.6%). This distribution more closely tracks the empirical impact hierarchy from RQ1, consistent with task relevance weighting in Stage 4. The performance gap between GPT models is consistent with answerability differences, as their relevance scores are similar (Table 9).

Training dynamics (Figure 6, Appendix) reveal a staged learning progression: non-redundancy is resolved first, followed by diversity, then answerability—consistent with the reward pipeline's implicit curriculum structure.

### 6.5. Qualitative Analysis

The quantitative results establish that the trained model asks fewer, more relevant questions. Table 10 shows representative clarification questions per model, illustrating recurring divergence patterns.

**Budget misallocation toward lower-impact categories.** Both baselines ask valid questions but concentrate their budget on moderate- and lower-impact categories (Table 9). GPT-5 Nano devotes three of four questions to version and reproduction information; GPT-5 focuses on configuration and role definitions. Our model leads with a screenshot request targeting Error Information—the highest-impact category—followed by a focused reproduction request and a single version question. All three models ask reasonable questions, but only ours is optimized to prioritize the most impactful information.

**Question burden and answerability.** GPT-5 Nano's questions are individually answerable but impose substantial cognitive load (e.g., Question 3 combines build workflow, Makefile usage, and the `.rst→.tex→.pdf` pipeline into a

single query). GPT-5's Question 3 asks users to distinguish explicit spaces from TeX glue—an internal rendering detail unlikely to be accessible to a typical reporter. All three of our model's questions involve concrete, observable artifacts (visual output, a code snippet, version numbers), reflecting the answerability strategies from RQ2 and illustrating how answerability training shapes not just *whether* questions can be answered, but *how easily*.

**Issue-conditioned vs. template-based questioning.** Baseline models tend toward reusable question structures that transfer broadly across issues—GPT-5 Nano's questions 1 and 3 could appear in any LaTeX-rendering bug report. Our model's questions directly reference the observable symptom (incorrect visual output) and the specific component (Python role highlighting), a direct consequence of the diversity reward penalizing cross-issue similarity in Stage 2.

**Strategic abstention.** In `matplotlib-26208`, GPT-5 Nano generates only redundant questions and GPT-5 only unanswerable ones; our model asks none. This abstention likely emerges as a byproduct of the staged reward design: when candidate questions would fail the reward constraints, silence is the better policy than imposing user burden without recovering useful information.

### 6.6. Reward Component Analysis

**Stage ablations.** To validate that each reward stage contributes meaningfully, we compare the full pipeline against the SFT checkpoint and two stage-ablated variants in Table 4. SFT alone reaches 24.2% (1.5 questions), confirming that supervised initialization establishes basic behavior without principled strategies—RL contributes +12.6 points. Removing Stage 3 (answerability) drops performance to 24.4%, confirming that unanswerable questions actively degrade performance through context pollution. Removing Stage 4 (task relevance) drops to 32.0%, validating the SHAP-weighted reward as a meaningful driver. The tension between answerability and relevance scores across ablations suggests the full pipeline learns a joint trade-off that single-objective optimization would fail to capture.

**Stage failure modes.** Each stage addresses a distinct failure mode earlier stages cannot catch. Without Stage 1, the model reframes stated information as open questions (e.g.,

given *the build fails with exit code 1*, asking *What exit code does the build return?*). Stage 2 catches convergence on reusable templates—broad version queries, generic reproduction requests—not caught by redundancy filtering alone. Stage 3 filters issue-specific but internally-targeted questions users cannot observe (e.g., *What is the internal execution order of the middleware stack?*). Stage 4 corrects frequency-importance misalignment from RQ1, preventing questions from clustering toward lower-impact categories. Each stage is thus a necessary precondition for the next. However, the model still struggles when underspecification requires deep code comprehension, identifying surface-level gaps but not bridging from symptoms to underlying implementation as reliably as GPT-5.

## 7. Related Work

### 7.1. Clarification and Ambiguity Resolution

Ambiguity handling in NLP typically decomposes into detection and clarification. Prior work on uncertainty estimation and self-disambiguation enables models to recognize when user input is incomplete or ambiguous (Lin et al., 2022; Wang et al., 2022; Hou et al., 2024). Pipeline-style systems combine detection with clarification generation (Tang et al., 2025; Zhang & Choi, 2025), often estimating the value of interaction as a proxy for detecting missing information (Zhang et al., 2025a; Zhang & Choi, 2025).

Benchmarks such as CLAMBER and ClarQ-LLM (Zhang et al., 2024; Gan et al., 2024) evaluate LLMs' ability to identify and resolve ambiguity across open-domain QA, search, and task-oriented dialog. Complementary works define taxonomies of linguistic, referential, and pragmatic ambiguity, measuring LLM robustness under multiple interpretations (Madge et al., 2025; Zhang et al., 2024; Niwa & Iso, 2024; Sumanathilaka et al., 2025). Multi-intent QA work (Niwa & Iso, 2024; Kim et al., 2024; Zhang & Choi, 2025) further highlights the value of targeted disambiguation. CQ generation techniques include framework-guided prompting (Mu et al., 2024; Tang et al., 2025) and ensemble-based selection (Zhang & Choi, 2025; Zhang et al., 2025a).

These systems generally optimize for conversational clarity or linguistic coverage focusing on singular or limited, linguistic points of ambiguity, whereas many real tasks require selecting multiple, most valuable piece of missing information. Our work focuses on this task-productivity perspective by identifying which clarification question yields the greatest downstream utility, under user answerability constraints, in complex underspecified problem settings.

### 7.2. Clarification in Agentic Settings

Ambiguity in agentic environments introduces additional challenges. Agents must decide when to query and what information blocks progress. Recent methods integrate struc-

tured uncertainty estimation to trigger interaction (Suri et al., 2025), and supervised or contrastive training to improve querying behavior (Zhang et al., 2025b; Chen et al., 2025), while other works analyze broader underspecification in SWE agents (Vijayvargiya et al., 2025). Other lines incorporate user modeling through Theory-of-Mind (Zhou et al., 2025), explicit classification–clarification modules (Darji & Lutellier, 2025), or proactive/personalized query policies optimized for user burden and task success (Sun et al., 2025). Software engineering is a frequent evaluation domain due to its rich specifications and verifiable success metrics (Jimenez et al., 2024).

Our work differs in two key ways. First, we isolate the single-turn clarification problem to study what makes clarifications maximally productive for downstream task completion, rather than modeling multi-turn interaction strategies. Second, we systematically quantify the impact of different information types on task success and use these empirical findings to design training objectives. Rather than relying on generic quality heuristics, we ground our approach in empirical analysis of which missing information matters most, which questions are answerable, and how these factors predict downstream performance.

## 8. Conclusions

In this work, we establish that effective clarification depends on two complementary properties: task relevance (which information impacts success) and user answerability (what users can realistically provide). Through Shapley analysis, we find a clear hierarchy of information needs that contribute to task success. Distributional analysis reveals four strategic characteristics to generating answerable questions: grounding in evidence, demanding specificity, minimizing scope, and ensuring actionability. Leveraging these insights, we train CLARITI, an 8B parameter module, matching GPT-5 while generating 41% fewer questions.

However, our study has limitations. Our single-turn design isolates clarification quality but does not capture multi-turn dynamics. The SWE focus for empirical grounding provides domain-specific findings, although our methodology generalizes. Finally, the LLM judge enables scalable evaluation, but may diverge from human judgment. Future work should implement the effective clarification strategies across domains and multi-turn settings. In conclusion, we establish a generalizable training methodology—quantify information impact, identify answerable characteristics, operationalize via multi-stage rewards—applicable wherever agents must extract information from users to complete tasks.

## Impact Statement

Our work focuses on improving how AI agents clarify underspecified tasks through better question generation. By enabling more efficient human-AI interaction, this research could reduce user burden and allow general models to improve performance in software engineering and similar technical domains. The techniques we develop are general-purpose methods and do not introduce novel risks beyond those present in large language model deployment, which have been documented in prior work.

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

# A. Appendix

## A.1. Information Needs Annotation

| Instance ID | Annotator Notes | Missing Information |
|---|---|---|
| 22 (astropy__astropy-13469) | The issue is stated in the description - When trying to convert a list of Tables to a NumPy array, it is automatically converted to the wrong data structure, but if the type is specified (dtype=object), a ValueError is raised. But since the referenced issue is a stackoverflow external link, it will be difficult to grasp exactly how the issue occurs, and what the expected outcome should be | Reproduction steps, Expected behavior |
| 78 (django__django-10531) | The description lacks details about the Django version affected (issue desc talks about some old versions of django), which is crucial since Django's handling of verbose names might differ across versions. Additionally, there is no mention of the specific Django admin components involved, such as whether the issue is with LogEntry objects, the ModelAdmin class, or specific methods that handle the rendering of change messages in the admin history. Without specifics on the Django components affected or a clearer outline of when the issue arises, a developer would face uncertainty in determining the exact scope and location of the necessary code changes. There is too much room for ambiguity. It is unclear what a successful solution would look like. | Version information, Expected behavior, Implementation details |
| 94 (django__django-11019) | The problem statement only explains the issue related to `MyForm().media`, which is that merging three media objects in Django form throws an unnecessary `MediaOrderConflictWarning` error. The error is also misleading. It suggests that "text-editor-extras.js" and "text-editor.js" are conflicting files, while the actual issue is the ordering of "color-picker.js" and "text-editor.js". Moreover, the desired solution is not mentioned in the description of how to solve this issue. | Error information, Expected behavior |
| 117 (django__django-11185) | The issue mentions that `Model.delete(keep_parents=True)` does not preserve all parent reverse relationships but does not specify which relationships are not preserved or provide examples of the failing cases. Without specific details or examples, it's unclear what exactly needs to be fixed. The phrase "relationships toward parents of parents, and so on" suggests a recursive or hierarchical problem, but it's not clear how deep this issue goes or what the expected behavior in various nested scenarios should be. This could lead to multiple interpretations of the problem. | Error information, Reproduction steps, Expected behavior |
| 129 (django__django-11279) | The problem statement requests a new functionality in a Django model structure that includes the `%(app_label)s` and `%(class)s` placeholder in the `name` argument for `BaseConstraint`, `CheckConstraint`, `UniqueConstraint`, and `Index`. The actual issue and error are not mentioned in the description. | Error information |

*Table 5.* Examples of underspecified instances from the SWE-bench dataset with annotator notes and identified missing information categories.

## A.2. Prompts for Controlled Underspecification Generation

**Prompt 1: Identifying Present Information Categories**

**System Message:** You are an expert at analyzing GitHub issues and identifying types of information present.
**User Message:**
Analyze the following GitHub issue and identify which categories of information are present.
TAXONOMY OF INFORMATION CATEGORIES:

- Error Information: What is going wrong, why is it wrong, and how do we know? (e.g., stack traces, error strings, incorrect

output descriptions, why is the behavior incorrect, any description of problematic behavior)

- Reproduction Steps: Under what conditions does the failure occur? (e.g., CLI commands, minimal inputs, trigger conditions)

- Implementation Details: How should the solution be implemented? (e.g., proposed solutions, implementation approaches)

- Version/Environment Information: What configuration is necessary? (e.g., dependency versions, OS details, config flags)

- External References: What external resources influence this? (e.g., API docs, datasets, upstream contracts, links, commit hashes, any description of what external reference contains)

- Expected Behavior: What should happen instead? (e.g., intended output format, correct return values, desired state, any description of correct behavior)

GITHUB ISSUE: [Issue text, test patch, and solution patch]
For each category that has information present in the issue, identify:

1. The category name (exactly as listed above)

2. Specific examples of that information from the issue

Output your analysis in JSON format with this structure:

```
{
    "Error Information": {
        "present": true/false,
        "examples": ["specific quote 1", "specific quote 2", ...]
    },
    ...
}
```

Be thorough and identify ALL categories that have relevant information. Include specific quotes/examples from the issue for each category marked as present. Output ONLY the JSON, no other text.

---

## Prompt 2: Generating Underspecified Issue Variants

**System Message:** You are a GitHub issue writer who creates realistic but incomplete issues.
**User Message:**
You are rewriting a GitHub issue to hide specific types of information while keeping it realistic.
Original GitHub Issue: [Issue text, test patch, and solution patch]
CATEGORY MAPPING (for reference - may have extra or missing items): [For each category to hide, examples of that information from the original issue]
You MUST completely remove ALL mentions of these information types: [List of categories to hide with their definitions and examples]
CRITICAL INSTRUCTIONS:

1. Remove EVERY mention of the information types listed above, except error information and expected behavior where if removing all mentions make the issue unnatural, then vaguely describe it, or remove important parts.

2. Use the category mapping as reference (but note it may have extra or missing items - use your judgment)

3. If you remove reproduction steps, remove sufficient/ALL commands, inputs, and trigger conditions

4. If you remove error information, remove sufficient stack traces, error messages, and incorrect output descriptions

5. If you remove implementation details, remove important/ALL proposed solutions and approaches

6. If you remove version/environment info, remove most/ALL dependency versions, OS details, configs

7. If you remove external references, remove ALL links, API docs, dataset mentions, commit hashes, and descriptions of external reference content.

8. If you remove expected behavior, remove sufficient descriptions of what should happen or correct behavior

9. Write like a REAL developer - natural, authentic, no theatrical language

10. Do NOT add extra formatting that real developers don't use

11. Do NOT mention what you removed or that information is missing

12. The result should sound like an incomplete but real issue

[If applicable: Previous rewrites hid these categories: ... Make sure your rewrite is DIFFERENT from these previous versions.]
Output ONLY the rewritten issue inside `<rewrite></rewrite>` tags. Include NO other text, explanations, or metadata.

## A.3. Annotation Process Example

This section illustrates the complete annotation pipeline for a single instance, showing how we transform original GitHub issues into underspecified versions through GPT-5-based annotation and selective information hiding.

**Instance ID**

`instance_id`: astropy__astropy-14182

**Step 1: Original Issue**

**Title:** Consider removing auto-transform of structured column into NdarrayMixin
**Description:**
Currently if you add a structured `np.array` to a Table, it gets turned into an `NdarrayMixin` (via the code below). While this mostly works, I am not sure this is necessary or desirable any more after #12644. Basically the original rational for `NdarrayMixin` was that structured dtype `Column` didn't quite work, in particular for serialization. So we pushed that out to a mixin class which would signal to unified I/O that it might not be supported.

```
# Structured ndarray gets viewed as a mixin unless already a valid
# mixin class
if (not isinstance(data, Column) and not data_is_mixin
        and isinstance(data, np.ndarray) and len(data.dtype) > 1):
    data = data.view(NdarrayMixin)
    data_is_mixin = True
```

**Proposal:**

- Add a FutureWarning here telling the user to wrap `data` in `Column` and that in the future (5.2) the structured array will be added as a `Column`.

- Change the behavior in 5.2 by removing this clause.

This is not critical for 5.1 but if we have the opportunity due to other (critical) bugfixes it might be nice to save 6 months in the change process.
cc: @mhvk

**Step 2: Categories Identification (GPT-5 Annotation)**

The following information categories were identified in the original issue:
**Error Information:** Not present
**Reproduction Steps:** Present

- Test directly adding various forms of structured ndarray columns to a table.

- `a = np.array([(1, 'a'), (2, 'b'), (3, 'c'), (4, 'd')], dtype='<i4,' + ('|U1'))`

- `t = Table([a], names=['a'])`

- `t['b'] = b`

**Implementation Details:** Present

- Add a FutureWarning telling the user to wrap `data` in `Column`

- Change the behavior in 5.2 by removing this clause

- Solution Patch removes the clause that views structured ndarray as NdarrayMixin

- Test Patch parameterizes test to compare behavior with and without NdarrayMixin view

**Version/Environment Information:** Present

- "in the future (5.2) the structured array will be added as a `Column`"

- "This is not critical for 5.1 but..."

**External References:** Present

- after #12644

- https://github.com/astropy/astropy/blob/main/CONTRIBUTING.md

**Expected Behavior:** Present

- In the future (5.2) the structured array will be added as a `Column`

- Add a FutureWarning telling users to wrap `data` in `Column`

---

### Step 3: Hidden Information Selection

For this instance, the following information types were randomly selected to be hidden:

1. **Version/Environment Information:** What configuration is necessary? (e.g., dependency versions, OS details, config flags)
   - Examples: "in the future (5.2) the structured array will be added as a `Column`"; "This is not critical for 5.1..."

2. **External References:** What external resources influence this? (e.g., API docs, datasets, upstream contracts, links, commit hashes)
   - Examples: "after #12644"; https://github.com/astropy/astropy/blob/main/CONTRIBUTING.md

3. **Expected Behavior:** What should happen instead? (e.g., intended output format, correct return values, desired state)
   - Examples: "in the future (5.2) the structured array will be added as a `Column`"; "Add a FutureWarning here telling the user to wrap `data` in `Column`"

---

### Step 4: Underspecified Issue (Rewritten with Hidden Information)

**Title:** Consider removing automatic wrapping of structured arrays in the tabular container
**Description:**
When a structured array is added to the tabular container, it is automatically wrapped in a mixin. This generally works, but the original rationale was related to limitations around handling structured data for serialization. With improved support for structured arrays, the automatic wrapping may no longer be necessary and can be surprising to users who expect a plain structured column.
This behavior also has downstream effects on type checks and I/O pathways, which can make the user experience inconsistent.
It would be useful to re-evaluate whether the auto-wrapping is still needed, and gather examples where it remains beneficial versus cases where a structured array behaves correctly without being treated as a mixin.
**Questions:**

- Are there current use cases where automatic wrapping is still required for correctness?

- Would removing the auto-wrapping introduce practical regressions for typical workflows?

- If changes are made, what level of notice or transition plan would be appropriate for users?

---

## A.4. Shapley analysis

We use an ensemble approach with three model architectures: (1) Logistic Regression with L2 regularization, (2) Random Forest with 100 trees and max depth 5, and (3) Gradient Boosting with 100 estimators and max depth 3. For each model, we compute SHAP values using appropriate explainers (LinearExplainer for logistic regression, TreeExplainer for tree-based models). We report the mean absolute SHAP value across all three models to provide a robust importance estimate that is not dependent on any single modeling choice. Cross-validation accuracy is computed using 5-fold stratified splits to verify

that models achieve reasonable predictive performance (all models achieve $> 0.80$ accuracy, substantially above the 0.52 baseline of predicting the majority class).

## A.5. Clarification Questions

> **Prompt: Generating Clarification Questions**
>
> **System Message:** You are an expert software developer reviewing a GitHub issue.
> **User Message:**
> You are an assistant helping to clarify software engineering issue descriptions. Given the following issue, generate a limited number of targeted clarification questions as a developer to get the information needed to solve the issue.
> Output format:
>
> ```
> 1. ...
> 2. ...
> 3. ...
> ```
>
> Problem Statement: [Underspecified issue text]

## A.6. Distributional Data Analysis Findings

Here we present the complete results from our D5 analysis comparing answerable versus non-answerable clarification questions. We conducted three analyses: (1) cross-model (pooling questions from all models), (2) GPT-5 within-model, and (3) GPT-Nano within-model. Each analysis identified characteristics that distinguish answerable questions (targeting information present in the original issue) from non-answerable questions (asking for unavailable information).

Tables 6–8 present the top 20 discoveries from each analysis. These discoveries are grouped by formulation strategy to emphasize that they describe *how to ask* (question design) rather than *what to ask about* (information categories from Table 1 in the main paper).

*Table 6.* Top 20 Question Formulation Characteristics: Cross-Model Analysis (All Models Pooled)

| Rank | Strategy | Characteristic | V' | Sig. |
|---|---|---|---|---|
| **Evidence Grounding** *(anchoring in concrete, verifiable artifacts)* | | | | |
| 1 | EG | Requests concrete, copy-pasteable evidence such as stack traces, logs, or minimal code | 0.191 | *** |
| 5 | EG | Anchors questions to concrete, user-observable artifacts like logs, warnings, and files | 0.111 | *** |
| 7 | EG | Requests exact error text and full stack traces to avoid paraphrase loss | 0.101 | *** |
| 9 | EG | Requests concrete code wiring details to reveal lifecycle and instantiation patterns | 0.095 | *** |
| 11 | EG | Focuses on observable behaviors such as error messages, stack traces, and logs | 0.079 | ** |
| 16 | EG | Ties questions to concrete artifacts the user can inspect directly | 0.045 | *** |
| **Precision Targeting** *(requesting exact values rather than categories)* | | | | |
| 3 | PT | Asks for exact API calls and option values to enable precise replication | 0.116 | *** |
| 6 | PT | References specific API calls and parameters the user likely executed | 0.108 | *** |
| 10 | PT | Uses method names and parameter signatures to ensure alignment on exact code path | 0.081 | *** |
| 17 | PT | Requests exact naming and import paths to resolve discovery/import ambiguity | 0.040 | |
| 20 | PT | Solicits context propagation paths to trace variables | 0.031 | ** |
| **Scope Minimization** *(isolating minimal reproducible cases)* | | | | |
| 4 | SM | Requests a minimal reproducible example with exact inputs, operations, and outputs | 0.116 | *** |
| 8 | SM | Prioritizes minimal reproducible examples to enable verification | 0.098 | *** |
| 12 | SM | Requests a minimal reproducible example to anchor discussion in concrete artifacts | 0.070 | *** |
| 13 | SM | Requests end-to-end snippets that show how data flows through the system | 0.060 | ** |
| 15 | SM | Isolates the smallest unit that can demonstrate the issue | 0.048 | ** |
| 18 | SM | Solicits input-output pairs to demonstrate the discrepancy explicitly | 0.038 | ** |
| **User Actionability** *(focusing on immediate, performable actions)* | | | | |
| 2 | UA | Limits questions to practical actions the user can perform immediately | 0.167 | *** |
| 14 | UA | Narrows the query to immediate, user-observable symptoms or artifacts | 0.057 | ** |
| 19 | UA | Asks for how constructs are built to track provenance | 0.033 | * |

*Strategy codes:* EG = Evidence Grounding, PT = Precision Targeting, SM = Scope Minimization, UA = User Actionability
*Significance:* *** $p < 0.001$, ** $p < 0.01$, * $p < 0.05$
*Note:* V' represents the difference in validator scores between answerable and non-answerable questions (higher values = more characteristic of answerable questions).

*Table 7.* Top 20 Question Formulation Characteristics: GPT-5 Within-Model Analysis

| Rank | Strategy | Characteristic | V' | Sig. |
|---|---|---|---|---|
| *Evidence Grounding* | | | | |
| 1 | EG | Prompts for copy-pasteable artifacts (commands run, config snippets, logs) | 0.178 | *** |
| 2 | EG | Focuses on immediate, user-observable artifacts | 0.164 | *** |
| 4 | EG | Asks for concrete, reproducible artifacts such as stack traces and code snippets | 0.149 | *** |
| 7 | EG | Solicits full stack traces and exact error strings to remove ambiguity | 0.113 | *** |
| 10 | EG | Aims for artifacts that can be validated or shared verbatim | 0.108 | *** |
| 15 | EG | Requests exception details or tracebacks during execution | 0.064 | * |
| *Precision Targeting* | | | | |
| 8 | PT | Frames around specific entry points or workflows the user actually executes | 0.111 | *** |
| 9 | PT | Specifies exact entry points rather than broad features | 0.110 | *** |
| 16 | PT | Emphasizes precise version and environment details | 0.061 | * |
| *Scope Minimization* | | | | |
| 3 | SM | Prioritizes minimal reproducible artifacts the user can generate | 0.160 | *** |
| 13 | SM | Seeks deterministic reproduction paths with ordered steps | 0.077 | * |
| 14 | SM | Asks for minimal reproducible examples without external tooling | 0.072 | ** |
| 18 | SM | Constrains scope to concrete behaviors of a single tool | 0.052 | * |
| 19 | SM | Requests direct mapping from inputs to outputs | 0.047 | * |
| *User Actionability* | | | | |
| 5 | UA | Targets information the user can readily inspect in their local repo or runtime | 0.134 | *** |
| 6 | UA | Seeks step-by-step details of what the user did and what happened next | 0.118 | *** |
| 12 | UA | Leverages standard diagnostics the user can run | 0.093 | ** |
| 20 | UA | Asks for measurable baselines and performance expectations | 0.047 | ** |
| *Other Strategies* | | | | |
| 11 | Other | Reduces ambiguity by anchoring in familiar workflows | 0.093 | ** |
| 17 | Other | Encourages confirmation of which known scenario occurred | 0.053 | |

*Strategy codes:* EG = Evidence Grounding, PT = Precision Targeting, SM = Scope Minimization, UA = User Actionability
*Significance:* *** $p < 0.001$, ** $p < 0.01$, * $p < 0.05$

*Table 8.* Top 20 Question Formulation Characteristics: GPT-Nano Within-Model Analysis

| Rank | Strategy | Characteristic | V' | Sig. |
|------|----------|----------------|-----|------|
| *Evidence Grounding* | | | | |
| 1 | EG | Frames requests around artifacts the user can share | 0.181 | *** |
| 2 | EG | Anchors on artifacts the user can directly observe or produce | 0.163 | *** |
| 5 | EG | Asks for exact identifiers and messages present in the user's environment | 0.097 | ** |
| 12 | EG | Requests before/after evidence to localize the discrepancy | 0.058 | *** |
| 13 | EG | Relies on stack traces and test outputs as ground truth | 0.050 | |
| 16 | EG | Minimizes ambiguity by asking for exact strings and snippets | 0.041 | * |
| 19 | EG | Requests complete contextual snippets around the problematic entry | 0.036 | |
| *Precision Targeting* | | | | |
| 6 | PT | Investigates how components are wired together in the user's code path | 0.066 | * |
| 8 | PT | Probes implementation wiring details the user can inspect | 0.062 | * |
| 9 | PT | Leverages terminology and structures likely present in the user's code | 0.062 | * |
| 15 | PT | Emphasizes exact locations to enable precise mapping of symptoms | 0.044 | |
| 18 | PT | Requests exact identifiers to disambiguate context | 0.036 | |
| *Scope Minimization* | | | | |
| 3 | SM | Invites a minimal reproducible example to ground discussion in code | 0.154 | *** |
| 4 | SM | Solicits a minimal reproducible example to anchor discussion | 0.129 | *** |
| 10 | SM | Asks for minimal reproducible examples to verify the issue | 0.059 | * |
| 14 | SM | Focuses on pipeline stage localization to isolate failure points | 0.044 | * |
| 17 | SM | Uses constrained alternatives to isolate causes | 0.041 | |
| 20 | SM | Narrows scope to a single component or interaction path | 0.032 | |
| *User Actionability* | | | | |
| 7 | UA | Ties questions to concrete tooling steps within a familiar workflow | 0.063 | * |
| 11 | UA | Aims to reproduce errors under controlled conditions in CI | 0.058 | |

*Strategy codes:* EG = Evidence Grounding, PT = Precision Targeting, SM = Scope Minimization, UA = User Actionability
*Significance:* *** $p < 0.001$, ** $p < 0.01$, * $p < 0.05$

Across all three analyses, we identified 60 top discoveries (20 per analysis). Table **??** summarizes the distribution of discoveries by formulation strategy and provides examples of how each strategy manifests across different analyses.

## A.7. Intrinsic Evaluations

---

**Prompt: Answerability Evaluation**

**System Message:** Evaluate whether each question can be answered from the original issue vs the underspecified issue. For EACH question, determine if it's answerable from each source. Respond ONLY with valid JSON.
**User Message:**
Underspecified Issue (Context): [Underspecified rewrite variant]
Original Issue: [Original problem statement]
Questions:

1. Question 1

2. Question 2

3. ...

For EACH question, determine:

1. Can it be answered from the ORIGINAL issue? (true/false)

2. Can it be answered from the UNDERSPECIFIED issue? (true/false)

A question that is generic like "provide more details", "clarify requirements" and does not ask for specific information should be marked as NOT answerable from either (both false).
Respond with ONLY this JSON (no extra text):

---

```
{
  "questions": [
    {
      "question_num": 1,
      "question_text": "<brief quote>",
      "answerable_from_original": true/false,
      "answerable_from_underspecified": true/false,
      "reasoning": "<brief explanation>"
    },
    ...
  ]
}
```

**Note:** Questions are categorized as: (1) *answerable_original_only* (answerable from original but not underspecified), (2) *non_answerable* (not answerable from either), or (3) *answerable_both* (answerable from both, discarded from analysis).

---

**Prompt: Task Relevance Evaluation**

**System Message:** You are evaluating whether clarification questions are relevant to solving a software engineering task. Respond ONLY with valid JSON.
**User Message:**
Task Context: [Underspecified issue rewrite variant]
Clarification Questions:

1. Question 1

2. Question 2

3. ...

For EACH question, evaluate whether it is relevant to solving the software engineering task described in the context.
A question is **relevant** if:

- It seeks information needed to understand, reproduce, or fix the issue

- It asks about technical details, error conditions, or implementation requirements

- The answer would help a developer make progress on the task

A question is **not relevant** if:

- It is overly generic (e.g., "Can you provide more details?")

- It asks about information unrelated to the technical problem

- It explores tangential topics not needed for the fix

Respond with ONLY this JSON (no extra text):

```
{
  "questions": [
    {
      "question_num": 1,
      "question_text": "<brief quote>",
      "is_relevant": true/false,
      "reasoning": "<brief explanation>"
    },
    ...
  ]
}
```

## A.8. Training Setup

### A.8.1. FOUR-STAGE REWARD PIPELINE

We design a four-stage reward pipeline that operationalizes task relevance (RQ1) and answerability (RQ2), plus two auxiliary criteria (non-redundancy and diversity) that prevent reward hacking. Each stage implements rejection filtering: candidates failing to meet a threshold at stage $t$ receive zero reward in subsequent stages, preventing the policy from gaming individual metrics.

**Stage 1: Non-redundancy reward.** Redundant questions waste user time by requesting information already available in the task description. We implement a two-pass evaluation for better reliability:

**Pass 1 (Answer extraction):** Qwen 3 32B (acting as judge) attempts to answer each generated question using only the underspecified issue description. Questions receive either: (a) substantive answer if information is present, or (b) *I don't know* if information is missing.

**Pass 2 (Redundancy classification):** The judge evaluates whether questions with substantive answers are genuinely redundant or represent legitimate partial information needs. The redundancy score is:

$$r_{\text{redundancy}} = 1 - \frac{\text{redundant\_count}}{\text{total\_questions}}$$

Generations with $r_{\text{redundancy}} < 0.5$ (more than half redundant) are filtered, receiving zero reward in subsequent stages.

**Stage 2: Diversity reward.** Generic questions like *Can you provide more details?* can pass Stage 1 (they are not redundant) but provide little value because they do not identify specific information needs. Further, models often identify "safe" questions that can get high rewards for different inputs. The diversity reward aims to penalize similarity between questions from same generation, and within different batches. Due to the technical nature of the questions, embedding-based approaches can lead to false positives/negatives. Thus, we again employ the judge model to identify similarities. High similarity across different issues indicates generic, non-specific questions. The specificity score is:

$$r_{\text{specificity}} = 1 - \frac{\text{similar\_count}}{\text{total\_questions}}$$

. This pushes the model toward issue-specific questions that reference concrete entities (file names, function names, error types) rather than vague requests. Generations with $r_{\text{specificity}} < 0.5$ are filtered.

**Stage 3: Answerability reward.** We operationalize RQ2's answerability criterion through a two-pass evaluation:

**Pass 1 (User knowledge simulation):** The judge receives the original fully-specified issue and attempts to answer each clarification question from it.

**Pass 2 (Answerability scoring):** The judge evaluates whether answers are substantive. The answerability score is:

$$r_{\text{answerability}} = \frac{\text{answerable\_count}}{\text{total\_questions}}$$

This ensures questions fall within realistic user knowledge bounds, implementing RQ2's finding. Stage 1 filtering is critical here as without it, redundant questions (which are trivially answerable from the full issue) would receive artificially high answerability scores, allowing the policy to game this reward.

**Stage 4: Task relevance reward.** We operationalize RQ1's importance hierarchy by assigning utility weights based on information categories. The judge classifies each question into our taxonomy categories or marks it irrelevant. Each category receives a weight proportional to its relative mean SHAP value from RQ1 (error information highest, external references lowest). The relevance score aggregates these weights:

$$r_{\text{relevance}} = \frac{1}{N} \sum_{i=1}^{N} w(\text{category}_i)$$

This directly optimizes for questions targeting high-impact information identified in RQ1. Stage 2's diversity filtering is crucial as without it, the policy could generate safe but similar, generic questions across issues that map to high-weight categories in principle but provide no practical value. The final reward is an equally weighted sum of the reward from each stage.

### A.8.2. MODEL ARCHITECTURE AND INFRASTRUCTURE

We train our clarification question generation model using Group Relative Policy Optimization (GRPO) starting from Qwen3-8B. The training infrastructure consists of three distributed components:

- **Actor model**: The policy model being optimized

- **Reference model**: Frozen copy of the initial policy for KL divergence computation

- **Reward model**: Qwen3-32B used for four-stage reward evaluation

We use AdamW with learning rate $5 \times 10^{-6}$, batch size 4, gradient accumulation of 4, weight decay 0.01, and $\epsilon = 10^{-8}$. During training, we generate $N = 8$ samples per prompt with temperature 0.9 and top-p sampling 0.9. Maximum sequence length is 1024 tokens for input and 256 tokens for generation. We use mean baseline for advantage computation and KL coefficient $\beta = 0.05$ to balance reward maximization with staying close to the reference policy.

---

**Stage 1: Redundancy Check - Pass 1 (Answer Questions)**

**System:** Answer questions based solely on provided context. If context doesn't contain enough information, respond with *I don't know*.
**User:**

```
Context:
[Task description]

Questions:
[Numbered list of questions]

Based ONLY on the context, answer each question. Format:
A1: <answer or "I don't know">
A2: <answer or "I don't know">
...
```

---

**Stage 1: Redundancy Check - Pass 2 (Evaluate Redundancy)**

**System:** Evaluate whether questions were already answered by context. Respond ONLY with valid JSON in the exact format shown below.
**User:**

```
Context:
[Task description]

Questions and Answers:
[Q-A pairs from Pass 1]

Count questions with SUFFICIENT answers (not "I don't know" or
"Formatting issue"). These are REDUNDANT.

Respond with ONLY this JSON (no extra text):
{"redundant_count": <number>, "total": <total>,
 "explanation": "<brief>"}
```

**Stage 2: Novelty Check**

**System:** Evaluate question novelty by comparing current questions to previous generations and questions within the same generation set. Respond ONLY with valid JSON.
**User:**

```
Previous Generations:
[Last 2 cached generations]

Current Generation:
[Current questions]

Count how many current questions are SIMILAR to previously asked questions or to other

questions in the current generation.

Respond with ONLY this JSON (no extra text):
{"similar_count": <number>, "total": <total>,
 "explanation": "<brief>"}
```

**Stage 3: Answerability Check - Pass 1 (Answer Questions from Original)**

**System:** Answer questions based solely on the original bug report. If the report doesn't contain enough information, respond with "I don't know". If a question is generic or doesn't ask for specific information, respond with "Generic question".
**User:**

```
Original Bug Report:
[Original complete issue]

Questions:
[Numbered list of questions]

Based ONLY on the original bug report, determine if the user
who filed the report can answer each question. For each question:
- If the report contains the information needed to answer:
  provide the answer
- If the report doesn't have this information: "I don't know"
- If the question is too generic ("provide more details", "clarify requirements"):

"Generic question"

Format:
A1: <answer or "I don't know" or "Generic question">
A2: <answer or "I don't know" or "Generic question">
...

Answers:
```

**Stage 3: Answerability Check - Pass 2 (Evaluate Answerability)**

**System:** Evaluate whether questions can be answered by the user based on their original bug report. Respond ONLY with valid JSON in the exact format shown below.
**User:**

```
Original Bug Report:
[Original complete issue]

Questions and Answers:
Q1: [question]
A1: [answer from Pass 1]
Q2: [question]
A2: [answer from Pass 1]
...
```

```
Count questions that are ANSWERABLE by the user (those with
actual answers, not "I don't know" or "Generic question").
Generic questions that don't ask for specific information are
NOT answerable.

Respond with ONLY this JSON (no extra text):
{"answerable_count": <number>, "total": <total>,
 "explanation": "<brief>"}

Your response:
```

**Stage 4: Utility Check**

**System:** You are classifying clarification questions by information type for software issues. Respond with valid JSON only.
**User:**

```
Task:
[Task description]

Questions:
[Current questions]

Classify EACH question by the information need it addresses:

INFORMATION TYPES:
- error_info: What is going wrong, why, and how?
- reproduction: Under what conditions does failure occur?
- expected_behavior: What should happen instead?
- implementation: How should the solution be implemented?
- external_refs: What external systems are relevant?
- version_env: What configuration is needed?
- irrelevant: Generic, off-topic, or mentions entities not in issue

Respond with ONLY this JSON:
{"classifications": [
    {"question_num": 1, "info_type": "<type>"},
    ...
]}
```

During training and inference, the policy model uses the following system prompt:

**Generation System Prompt**

Generate questions, if any, to ask the user to recover missing information required to solve the task.

### A.8.3. TRAINING REWARD CURVES

Figure 5 shows the progression of rewards across training. The mean reward increases from approximately 0.2 to 0.6 over 200 training steps, demonstrating that the model successfully learns to generate higher-quality clarification questions through the GRPO objective.

Figure 6 breaks down the contribution of each reward stage.

### A.9. Information Needs in CQs

### A.10. Qualitative Examples

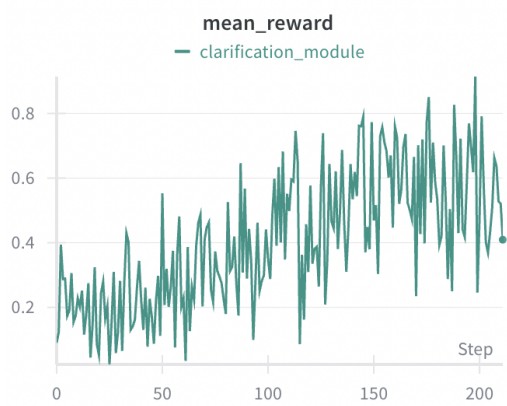

*Figure 5.* Mean reward progression during GRPO training. The reward increases steadily, indicating successful policy optimization toward generating non-redundant, novel, answerable, and useful clarification questions.

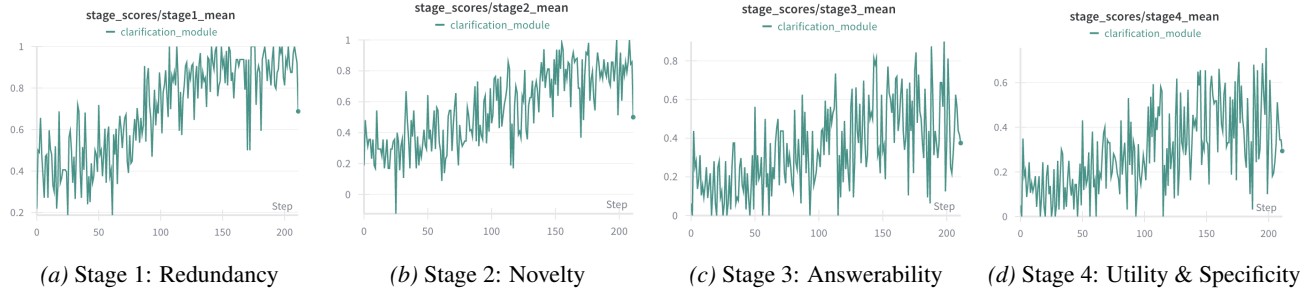

*(a)* Stage 1: Redundancy      *(b)* Stage 2: Novelty      *(c)* Stage 3: Answerability      *(d)* Stage 4: Utility & Specificity

*Figure 6.* Stage-wise reward progression during training. Each stage captures a different quality dimension of clarification questions, and the combined optimization across all stages drives the policy toward high-quality question generation.

*Table 9.* We compare the distribution of questions produced by different models with the empirical impact of each question categories on task success (from RQ1 in §4). Our model's distribution more closely tracks the impact hierarchy than either baseline.

| Category | Weight | Nano | GPT-5 | CLARITI |
|---|---|---|---|---|
| Error Info | 0.340 | 6.0% | 10.2% | 26.4% |
| Impl. Details | 0.184 | 29.9% | 16.4% | 12.3% |
| Version/Env. | 0.163 | 11.8% | 19.5% | 16.1% |
| Reproduction | 0.143 | 24.7% | 30.2% | 19.6% |
| Expected Behav. | 0.105 | 24.1% | 21.5% | 16.7% |
| External Ref. | 0.065 | 3.4% | 2.2% | 5.5% |

*Table 10.* Qualitative Analysis of Clarification Questions Across Models. Questions abridged due to space limitations

| Instance | GPT-5 Nano | GPT-5 | Ours |
|---|---|---|---|
| sphinx-10435 | 1. Exact versions of Sphinx, Python, Pygments, LaTeX toolchain? Any custom `latex_elements` or inline role? *[A, ver]*

2. Provide minimal RST snippet reproducing the issue with the inline role (e.g., `:python:'...').` Do spaces occur without surrounding text? *[R, repro]*

3. Exact PDF build workflow? Commands, Makefile, latexmk usage, and .rst→.tex→.pdf sequence? Custom scripts? *[A, repro]*

4. Limited to Python role or occurs with other languages/roles and block highlights? Spaces in HTML output? *[U, repro]* | 1. Versions (Sphinx, Pygments, TeX engine, OS) and build command for PDF? *[A, ver]*

2. How is the role defined? Share `conf.py` config and RST snippet. *[A, ver]*

3. Minimal project (one .rst + conf.py) and generated .tex snippet? Explicit spaces or TeX glue in .tex? *[U, repro]*

4. When did this start? Which versions didn't have the issue? *[A, ver]* | 1. Screenshots showing correct vs incorrect output? *[A, err]*

2. Reproducible example with Python role highlighting? *[A, repro]*

3. Relevant software versions? *[A, ver]* |
| matplotlib-26208 | 1. Does replacing string x-axis with numeric (e.g., [0,1]) or datetime reproduce? Tied to string labels or stacking/axis interaction? *[R, repro]*

2. Besides tested orders, tried other combinations (ax1.plot + ax2.stackplot, both on same axis)? Results? *[R, repro]*

3. Does this occur with different backends (Qt5Agg, TkAgg) or specific to module backend? *[U, ver]* | 1. Affects only metadata (ax1.dataLim) or rendered result? What are ax1.get_ybound(), ax1.get_ylim() after print? Figure display correct? *[U, err]*

2. Reproduce with numeric/datetime x data instead of strings? Replace df1_index with [0,1] and share dataLim. *[U, repro]*

3. Do ax1.relim() and ax1.autoscale_view() restore finite dataLims? Calling before/after ax2 plotting? *[U, repro]* | 1. N/A |
| django-14373 | 1. Exact year formatting token convention (YYYY/yyyy always 4-digit zero-padded, YY exactly 2 digits)? Other width-affecting tokens? *[A, exp]*

2. How to handle edge years: BCE/astronomical (0000, -0001), large years (10000+)? Fixed width or allow wider? *[U, exp]*

3. Provide failing test cases: input dates, timezones, format strings, expected vs actual, locale/calendar? *[U, repro]* | 1. Share examples (date/time, format string, current vs expected) for years 0–9, 10–99, 100, negative, 10000? *[U, repro]*

2. Intended year field convention: fixed/variable width, signed for BCE, how to represent outside 0000–9999? Align with ISO 8601? *[A, exp]*

3. Locale/timezone considerations? Tests assume fixed locale/timezone? *[U, ver]*

4. Environments showing discrepancy? Backward-compatibility constraints? *[U, ver]* | 1. How does the issue show inconsistency between expected (0-padded) and actual (no padding)? *[A, err]*

2. Which components (YearFormat, YearDeltaFormat) are affected? *[A, err]* |

**Legend:** A=Answerable, U=Unanswerable, R=Redundant. **Types:** err=error_info, repro=reproduction, exp=expected_behavior, ver=version_env.

