# OpenReview forum: "Asking What Matters: Reward-Driven Clarification for Software Engineering Tasks"
_ICML.cc/2026/Conference — ICML 2026 regular_

### Official Review · Reviewer_cYqX · 2026-02-25

**Soundness:** 3
**Presentation:** 3
**Significance:** 4
**Originality:** 4
**Overall Recommendation:** 5
**Confidence:** 3

**Summary:**

This paper investigates how to teach AI agents (specifically in Software Engineering contexts) to ask effective clarification questions when user instructions are underspecified. The authors argue that not all missing information is equally valuable and that questions must be "answerable" by a typical user.

**Compliance With Llm Reviewing Policy:**

Affirmed.

**Key Questions For Authors:**

- In your evaluation (RQ3), you simulate user responses by extracting information from the fully specified issue. How does this account for the "Unknown Unknowns"? If a real user omitted "Error Information" because they didn't know how to find the logs, your simulation assumes they can provide it if asked. Does this inflate the performance of the clarification module?

- Why were existing ambiguity-resolution baselines (e.g., CLAMBER or uncertainty-based querying methods mentioned in Related Work) not directly compared in Table 4? Comparing against GPT-5 is a "model" comparison, not a "method" comparison.

**Limitations:**

yes

**Strengths And Weaknesses:**

# Soundness: 3 (Good)

## Strengths
- The use of Shapley values (RQ1) to quantify the marginal contribution of different information types is a statistically sound approach to feature importance, moving beyond simple qualitative heuristics.
- The experimental design for RQ3 cleanly measures the utility of the questions themselves.
- The use of "controlled underspecification" creates a solid ground-truth for evaluation.

## Weaknesses
- The reliance on simulated users (extracting answers from the original fully-specified issue) is a standard but limiting proxy. While the authors claim questions are "answerable by typical users," the simulation assumes the user has the information present in the original issue text. In reality, users often underspecify because they don't know that information, not just because they forgot to write it down. The paper acknowledges this in RQ2 but the simulation in RQ3 might still be overly optimistic about what a user can provide.
- The baseline comparison includes "No clarification," "Nano," "GPT-5," and "Full issue," but lacks a strong baseline of an existing clarification method (e.g., a standard entropy-based or uncertainty-based active learning baseline, or a simple prompt-engineered Llama-3-70B).

# Presentation: 3 (Good)

## Strengths
- The writing is clear, logical, and easy to follow.

## Weaknesses
- The description of the Shapley value calculation (Section 2.3) is slightly high-level in the main text; ensuring the specific feature vectors and model used for this calculation are reproducible is critical.
- The "distributional analysis" in RQ2 mentions "Vargha-Delaney V'" and "four strategic themes," but more concrete examples of the 60 unique characteristics discovered would effectively ground the abstract themes.


# Significance: 4 (Excellent)

## Strengths
- Most current agent research focuses on planning or code generation given a perfect prompt. This paper targets the realistic scenario where prompts are imperfect.
- The demonstration that an 8B model can reach parity with GPT-5 levels of utility through specialized training suggests a path toward efficient, on-device agentic assistants.

# Originality: 4 (Excellent)
## Strengths
- The application of Shapley values to rank information types in prompts is a novel contribution to the specific domain of SWE agents.
- The explicit training signal based on "answerability" (avoiding questions the user likely cannot answer) addresses a specific pain point in human-agent interaction that is often overlooked in favor of maximizing information gain regardless of user burden.
- This work tackles task-oriented underspecification (missing requirements/context), which is more relevant for complex problem solving.

---

> ### Author Rebuttal · Authors · 2026-03-31
>
> Thank you for the thorough and positive assessment. We address your questions below.
>
> > Simulated user is an upper-bound for performance
>
> We agree this is a nuanced point. For unknown unknowns specifically (cases where users underspecify because they lack the information entirely) our simulation is optimistic, as it assumes the user can provide what appears in the original issue. However, this is precisely what motivates our answerability reward: RQ2's analysis identifies question characteristics that users systematically cannot answer, and Stage 3 of our training pipeline explicitly penalizes such questions. So while the simulation may overestimate absolute performance, the answerability component of our framework directly targets the failure mode identified. We acknowledge this as a limitation of working within the existing ecosystem as there are currently no datasets of real underspecified software engineering issues paired with user responses, which makes simulation necessary at this stage. Creating such a dataset and conducting real-user evaluation is a direction we are very interested in pursuing as direct future work.
>
> > Method-level baselines
>
> Existing clarification methods such as CLAMBER and uncertainty-based querying target single-point linguistic ambiguity in open-domain QA and do not directly transfer to the multi-gap, task-oriented underspecification setting we study. More broadly, our framework is an orthogonal contribution as any approach that generates clarification questions could be further improved by optimizing for task relevance and answerability, whether via prompting, or explicit training. To further strengthen our empirical claims we include ablations demonstrating the contribution of each reward stage:
>
> | Ablation | Success ↑ | Answerability ↑ | Relevance ↑ | #Qs ↓ |
> |---|---|---|---|---|
> | No clarification | 22.40 | — | — | — |
> | SFT | 24.20 | .382 | 0.583 | 1.5 |
> | Full pipeline (Ours) | 36.80 | .373 | .622 | 3.0 |
> | w/o Stage 3 (Answerability) | 24.40 | .341 | .613 | 2.5 |
> | w/o Stage 4 (Task relevance) | 32.00 | .390 | .596 | 2.2 |
> | Fully specified | 41.60 | — | — | — |
>
> The ablations reveal several key insights. Removing Stage 3 (answerability) drops task success to 24.40% which is barely above no clarification (22.40%), confirming that unanswerable questions actively degrade performance by injecting noise rather than being merely neutral. Removing Stage 4 (task relevance) produces a drop from 36.80% to 32.00%, validating that SHAP-weighted reward is a key driver of task success gains. SFT alone reaches 24.20%, showing RL training adds substantial gains. Finally, there exists a tension between answerability and relevance scores across ablations suggesting the full pipeline learns a joint trade-off rather than independently maximizing either dimension, which single-objective optimization would fail to capture.
>
> Additional details on the Shapley analysis and the top 20 unique answerability characteristics are provided in Appendices A.4 and A.6 respectively and we will add more detail in the revision.

---

> > ### Author Rebuttal · Reviewer_cYqX · 2026-04-04
> >
> > Thank you for your rebuttal. After carefully considering your points as well as the feedback from other reviewers, I believe the score I have given is fair and reasonable.

---

### Official Review · Reviewer_7BNz · 2026-02-26

**Soundness:** 2
**Presentation:** 3
**Significance:** 4
**Originality:** 4
**Overall Recommendation:** 4
**Confidence:** 3

**Summary:**

This paper studies how software engineering agents should ask clarification questions when user requests are underspecified. It argues that effective clarification depends on two key properties: **task relevance**, meaning the question targets information that most improves downstream success, and **user answerability**, meaning the question asks for information that a typical user can realistically provide. To support this, the paper builds a taxonomy of missing information in SWE-Bench issues, analyzes which information types matter most for task success, studies what makes clarification questions answerable, and then trains a clarification module with reward signals based on these insights. In end-to-end evaluation, the proposed model achieves performance comparable to a strong GPT-5 baseline while asking fewer questions, suggesting better questioning efficiency.

**Compliance With Llm Reviewing Policy:**

Affirmed.

**Final Justification:**

The rebuttal addressed my main concerns

**Key Questions For Authors:**

see weaknesses

**Limitations:**

yes

**Strengths And Weaknesses:**

## Strengths

1. **Clear and practical problem formulation.** The paper focuses on an important and realistic issue for agent systems: asking useful clarification questions under underspecified instructions.

2.  **Strong conceptual decomposition.** The distinction between **task relevance** and **user answerability** is intuitive, well-motivated, and provides a clean framework for thinking about clarification quality.

3. **Useful practical insight.** The finding that asking more questions is not always better is valuable. The paper shows that too many low-quality or unanswerable questions can hurt performance, which is an important lesson for interactive agents.

4.  **Relevant benchmark and objective evaluation.** Using SWE-Bench-style tasks gives the work a concrete downstream metric rather than relying only on subjective judgments of question quality.

## Weaknesses

1. Limited baselines and insufficient experimental validation. The experimental comparison is not fully convincing because the baseline set is narrow. The most important missing baseline is an RL-based variant that uses the same training framework but removes the proposed derived rewards, like Task Relevance and Answerability. Without such a control, it is difficult to determine whether the gains come from the specific reward design or simply from applying RL fine-tuning. More broadly, the empirical section would be stronger with deeper ablations and more robust sensitivity analyses. Alternatively, maybe just prompt the model to rank the generated clarification questions with your derived metrics can show improvements, which would also be convincing experiment to add.

2. GRPO Training dynamic seems to be quite unstable as shown in Figure 2,3 in Appendix.

3. Heavy reliance on synthetic supervision and evaluation. A substantial part of the pipeline depends on LLM-generated or LLM-judged signals: underspecified issue rewrites are synthetically constructed, question answerability is judged by GPT-5, and user answers are simulated from the original issue text. This makes the setup scalable, but it also introduces the risk that the method is optimized for the assumptions and biases of the same LLM-based pipeline rather than for real human interactions. Some human evaluation might be helpful to validate.

4. SHAP may look more precise than the setup justifies.
They use SHAP to rank missing-information categories by contribution to task success, but the input is a very coarse binary presence/absence vector over only six hand-designed categories. That means the result depends heavily on the taxonomy itself and ignores within-category variation. For example, one “Error Information” omission could be a trivial warning, another could be a decisive stack trace, yet both are treated identically.

While I appreciate this paper as highly timely and novel, the experiments are not very convincing. I think authors could benefit from conducting more experiments to show effectiveness and robustness and more carefully consider the position and thus the structure of the paper (analysis vs method).

---

> ### Author Rebuttal · Authors · 2026-03-31
>
> Thank you for the generous assessment of originality and significance, and we address each weakness below.
>
> > Baselines and ablations
>
> We present the complete set of updated results below that demonstrate the impact of each reward stage:
>
> | Ablation | Success (%) ↑ | Answerability ↑ | Relevance ↑ | #Qs ↓ |
> |---|---|---|---|---|
> | No clarification | 22.40 | — | — | — |
> | SFT | 24.20 | .382 | 0.583 | 1.5 |
> | Full pipeline (Ours) | 36.80 | .373 | .622 | 3.0 |
> | w/o Stage 3 (Answerability) | 24.40 | .341 | .613 | 2.5 |
> | w/o Stage 4 (Task relevance) | 32.00 | .390 | .596 | 2.2 |
> | Fully specified | 41.60 | — | — | — |
>
> Each stage contributes meaningfully. Removing Stage 3 (answerability) substantially drops task success to 24.40%, barely above no clarification (22.40%), confirming that unanswerable questions actively degrade performance by injecting noise. Qualitative analysis further reveals this absence enables reward hacking behavior to satisfy Stage 4, directly addressing the concern of whether gains stem from reward design or RL fine-tuning alone.
>
> Removing Stage 4 (task relevance) produces a drop from 36.80% to 32.00%, validating the SHAP-weighted reward as a key driver of task success gains. SFT alone reaches 24.20%, showing RL training adds substantial gains (+12.6 points) beyond supervised initialization. Notably, SFT asks the fewest questions (1.5) and achieves the lowest task success among variants, suggesting the model learns not just to ask better questions but the right number. The tension between answerability and relevance scores across ablations further suggests the full pipeline learns a joint trade-off that single-objective optimization would fail to capture.
>
>
> > Baseline of prompting a model to rank generated questions using derived metrics
>
> This is an interesting suggestion. However, we note that our reward signals are not straightforwardly applicable as inference-time ranking criteria. Answerability in particular requires simulating user responses, which at inference time means actually asking the user. Our trained model internalizes these preferences generatively, eliminating the need for inference-time filtering entirely. That said, encoding our reward design into a prompt-based judge for ranking is an interesting direction we leave to future work.
>
> > GRPO training instability
>
> We acknowledge the noisy reward curves. This is expected given the difficulty of the reward signal and the small judge models used during training. Critically, the instability does not affect reliability of the final model: across multiple training runs with different configurations (varying learning rates, KL coefficients, and stage transitions), we observe generally consistent final task success and answerability scores. Other GRPO/RL works also observe this phenomenon (Zheng, Chujie, et al. "Group Sequence Policy Optimization." arXiv, 2025). We will present these alongside smoothed curves in the final version for easier reading.
>
> > Synthetic supervision
>
> We acknowledge this as a limitation which helps scale interaction-based tasks. Several design choices mitigate circularity. Most importantly, the final task success metric is fully objective (test case pass/fail). Additionally, underspecified rewrites are manually validated on 50 held-out issues, and training data uses DeepSeek-V3 while all evaluation judges use GPT-5, preventing self-reinforcing model-specific biases. Real human evaluation remains an important direction for future work.
>
> > SHAP coarseness
>
> We agree that within-category variation exists — an "Error Information" omission could range from a trivial warning to a decisive stack trace. However, the coarse-grained signal is sufficient for our purpose: we use SHAP weights to set relative reward scaling across categories, not to rank individual instances. Importantly, the binary presence/absence annotation scheme is a deliberate choice that improves annotation consistency as fine-grained within-category distinctions would introduce substantial annotator disagreement and noise, yielding less consistent SHAP estimates. Even with within-category variation, if Error Information systematically contributes more to task success than Expected Behavior across the dataset, the between-category signal is actionable for reward design. We treat finer-grained attribution as a natural extension and will make this distinction explicit in the paper.
>
> > Paper structure (analysis vs method)
>
> We will restructure the paper to more explicitly position RQ1 and RQ2 as motivating analysis and RQ3 as the methodological contribution, making the analysis-to-method pipeline clearer throughout.

---

> > ### Author Rebuttal · Reviewer_7BNz · 2026-04-03
> >
> > The rebuttal addressed my main concern. I raised my score to 4.

---

### Official Review · Reviewer_Jvf1 · 2026-03-13

**Soundness:** 2
**Presentation:** 3
**Significance:** 3
**Originality:** 3
**Overall Recommendation:** 4
**Confidence:** 4

**Summary:**

The authors propose an empirically-motivated RL pipeline to train LLM agents to ask better clarification questions in the context of a software engineering task. Specifically, the authors first manually produce a taxonomy of information needs in SWE Bench issues that were marked by expert annotators as underspecified and then determine the importance of each information need for downstream task success through Shapley value analysis. The authors separately conduct a study of what makes clarification questions answerable by users. Using the insights on question importance and answerability, the authors propose a 4-stage RL reward formulation, which they use to train Qwen3-8B to generate clarification questions. With this pipeline, Qwen3-8B generates clarification questions leading to similar downstream accuracy as GPT-5-generated questions while being more efficient.

**Compliance With Llm Reviewing Policy:**

Affirmed.

**Final Justification:**

I raised my score to a 4 in response to the ablation experiments. I think the thoroughness of the data analysis and novelty of the empirically-motivated reward formulation outweigh the limitations of the method -- namely, its reliance on expert annotation. I believe a comparison against outcome-based rewards would have helped further contextualize the performance of the method (which is why the score is a 4 rather than a 5), although I understand the authors' arguments against including it and recognize that this may be computationally infeasible during the rebuttal period.

**Key Questions For Authors:**

- W1: How much dependence does the method have on the presence of expert annotations detailing the particular kinds of information needs for a given domain? How burdensome is the human annotation process and could this be automated with an LLM?
- W2: Could the authors provide an ablation study for their proposed four-stage reward?
- W3-A: What is the performance of the SFT initialization?
- W3-B: How necessary is the task relevance reward? If I understand correctly, this is the only part of the reward formulation that explicitly depends on domain-specific information. Could we obtain similar clarification question quality by instead naively optimizing on the downstream performance of the coding agent that conditions on the results of the clarification questions?

I do find the task setting to be distinctive and the focus on data to be a strong positive. I will increase my score if the questions above are answered.

**Limitations:**

Yes.

**Strengths And Weaknesses:**

There are many positives to the paper:
- The authors focus on the important question of training LLMs to generate clarification questions. Particularly, unlike many papers with this goal, the task setting they study has a large search space of possible questions and is highly realistic, being built on top of real-world user data from SWE Bench.
- The paper's heavy emphasis on converting insights from a thorough analysis of data into methodological details is a breath of fresh air when many such papers often directly move towards an RL algorithm with little concern on the underlying task or domain.
- The method results in clear gains in performance. Particularly, by training the model to only produce clarification questions and using a reward model that assigns different importance to questions targeting different information needs, the authors are able to perform RL training without needing to additionally run a coding agent.

However, I think that there are also numerous important flaws:
- W1: The key flaw is that the task relevance reward assumes expensive human analysis of existing data to categorize the information needs for a specific, given task. This is exacerbated by the fact that the task relevances themselves are dependent on the performance of a particular coding agent. This problem would be largely alleviated if the annotation process itself could be automated.
- W2: There are no ablations that validate whether the proposed four-stage reward pipeline is necessary empirically.
- W3: The experiments also miss results from two baselines. One, it would be helpful to place the performance of the initial SFT checkpoint to contextualize how much RL training improves model behavior. Two, I am curious as to how a naive approach that performs RL training on the SFT checkpoint with a reward that is solely concerned with downstream task success would perform.

---

> ### Author Rebuttal · Authors · 2026-03-31
>
> Thank you for the detailed feedback. We are encouraged that you find the task setting distinctive and the data-driven focus a strong positive. We hope that the following experiments and clarifications about our setup address your concerns.
>
> > Expert annotation dependence and automation
>
> Our framework requires only a coarse category taxonomy, not fine-grained per-instance annotation. Identifying broad categories (e.g., our 6) requires experts analyzing a small set of representative tasks (30 in this work). This is a one-time, lightweight effort analogous to standard codebook development (MacQueen et al., 1998). Current LLMs lack the broad knowledge of task types to create the categorization and expert input here is highly beneficial. Critically, there is no expert annotation required at the instance level or agent level. All subsequent steps — per-instance category annotation, SHAP weight computation, and reward — can be fully automated via LLM judges, as already done in RQ1 and in RQ3 Stage 4. We will add an explicit discussion of this in the paper.
>
> > Reward pipeline ablations and SFT checkpoint performance
>
> Thank you for suggesting this important experiment. We present the complete set of updated results below that we will incorporate in the paper:
>
> | Ablation | Success ↑ | Answerability ↑ | Relevance ↑ | #Qs ↓ |
> |---|---|---|---|---|
> | No clarification | 22.40 | — | — | — |
> | SFT | 24.20 | .382 | 0.583 | 1.5 |
> | Full pipeline (Ours) | 36.80 | .373 | .622 | 3.0 |
> | w/o Stage 3 (Answerability) | 24.40 | .341 | .613 | 2.5 |
> | w/o Stage 4 (Task relevance) | 32.00 | .390 | .596 | 2.2 |
> | Fully specified | 41.60% | — | — | — |
>
> Each stage contributes meaningfully. Removing Stage 3 (answerability) substantially drops task success to 24.40%, barely above no clarification (22.40%), confirming that unanswerable questions actively degrade performance by injecting noise. Qualitative analysis further reveals this absence enables reward hacking behavior to satisfy Stage 4.
>
> Removing Stage 4 (task relevance) produces a drop from 36.80% to 32.00%, validating the SHAP-weighted reward as a key driver of task success gains. SFT alone reaches 24.20%, showing RL training adds substantial gains (+12.6 points) beyond supervised initialization. Notably, SFT asks the fewest questions (1.5) yet achieves the lowest task success among variants, suggesting the model learns not just to ask better questions but the right number.
>
> The tension between answerability and relevance scores across ablations further suggests the full pipeline learns a joint trade-off that single-objective optimization would fail to capture. We will incorporate these results into the paper.
>
> > Outcome-based RL as reward
>
> Directly using downstream coding agent success as a training reward is precisely the bottleneck our work is designed to circumvent. Outcome-based RL would require running full agent trajectories per training step — dozens of LLM calls over tens of thousands of tokens each — compared to our reward which requires only a few LLM judge queries of hundreds of tokens per question. Clarification quality becomes a black box, with no signal distinguishing *why* clarification helped or failed. Our core contribution is decomposing this into intrinsic, verifiable proxies to identify what makes clarification effective: RQ1's Shapley analysis empirically validates that information category presence predicts task success, grounding task relevance as a principled proxy for outcome, while answerability captures user burden that outcome-based RL would need to discover implicitly, if at all. We will make this framing more explicit in the paper, positioning outcome-based RL as the intractable objective our method is designed to circumvent.

---

> > ### Author Rebuttal · Reviewer_Jvf1 · 2026-04-02
> >
> > I thank the authors for their rebuttal, which resolves most of my concerns. I'm raising my score up to a 4 in response to the ablation experiments. I think the thoroughness of the data analysis and novelty of the empirically-motivated reward formulation outweigh the limitations of the method -- namely, its reliance on expert annotation. I believe a comparison against outcome-based rewards would have helped further contextualize the performance of the method, although I understand the authors' arguments against including it and recognize that this may be computationally infeasible during the rebuttal period.

---

### Official Review · Reviewer_KiVJ · 2026-03-19

**Soundness:** 3
**Presentation:** 4
**Significance:** 3
**Originality:** 3
**Overall Recommendation:** 4
**Confidence:** 4

**Summary:**

The work studies how agents should ask clarification questions in underspecified SWE agent tasks.

First, by manual inspection of 500 SWE-bench issues, the authors create a taxonomy of 6 broad categories of missing information that cover the majority of the cases of underspecification in the real world. Using Shapley value analysis, they show that the marginal impact of each category on downstream task success is different, with missing error information being the most impactful one.

To enable controlled experiments, they further perturb each task instruction to create an underspecified variant of each type of missing information (done via prompting LLM).

They then study the answerability of questions by prompt GPTs to generate clarification questions and label them as user answerable or not by GPT-5, and manually analyze them. They identify 4 key characteristics that distinguish answerable from unanswerable questions, and show that unanswerable questions degrade performance by adding noise.

Finally, they design an SFT + RL pipeline to train a clarification model (Qwen-3 8B) that optimizes for both task relevance and answerability, and evaluate it on underspecified issues. The trained model achieves 90% of fully-specified task performance while asking significantly fewer questions than GPT-5.

**Compliance With Llm Reviewing Policy:**

Affirmed.

**Key Questions For Authors:**

--

**Limitations:**

yes

**Strengths And Weaknesses:**

# Strengths

* Very detailed qualitative and quantitative analysis of asking clarifications in SWE tasks.
  The resulting taxonomy is insightful and can be useful for future research.
* RL reward metrics are well-designed and well-motivated to target the key qualities of good clarification questions.
* The trained model achieves good performance while asking fewer questions.
* The paper is clearly written and easy to follow.

# Weaknesses

The entire clarification model evaluation is only based on 250 examples, which is a relatively small set, to make strong claims. Additionally, all of these tasks come from SWE-Bench. Since the whole analysis was also done on SWE-Bench tasks, it would strengthen the paper to evaluate on other SWE task benchmarks as well, like SWE-Bench Pro, SWE-Bench+, etc. It would show whether the findings and trained model generalizes to other task distributions.

---

> ### Author Rebuttal · Authors · 2026-03-31
>
> Thank you for the positive assessment of the analysis on clarification and its impact on future research. We address the concern about evaluation generalization by clarifying our setup and conducting additional experiments to make our claim stronger.
>
> > The current evaluation set of 250 issues is small to make strong claims. Evaluating on other SWE benchmarks can strengthen claim of generalization
>
> Our training and evaluation distributions are already meaningfully different as we train on SWE-Gym Raw (Pan et al., 2024) which contains a broader, and different set of raw GitHub repositories than SWE-Bench Verified. Further, to prevent model biases from playing a role, training data is generated with DeepSeek-V3 while all evaluation judges and datasets use GPT-5.
>
> To further evaluate generalization to new distributions and provide more evidence for our model’s performance, we additionally benchmark on SWE-Bench Live (Lite subset, 100 instances). We select this dataset because it exhibits a wider performance range for open-source models, enabling more informative comparisons than alternatives such as SWE-Bench Pro, where uniformly low scores prevent meaningful differences.
>
> Full task success results require infrastructure changes to support the new benchmark and will be incorporated into the revision due to time constraints. However, we report intrinsic metric scores on SWE-Bench Live as an initial analysis, which only require LLM judge inference rather than full agent rollouts and evaluation:
>
> | Model | Answerability | Task relevance
> |---|---|---|
> | GPT-5 Nano | 0.324 | 0.604 |
> | GPT-5 | 0.229 | 0.607 |
> | Ours | 0.334 | 0.672 |
>
> Our model shows strong answerability and relevance scores on the new distribution, suggesting that the question characteristics learned during training generalize beyond SWE-Bench Verified. Critically, our ablations show that these intrinsic metrics are strong predictors of task success — removing either stage causes substantial performance drops — providing principled grounds to expect that metric-level generalization implies task success generalization.

---

### Decision · Program_Chairs · 2026-04-30

**Decision:**

Accept (regular)

**Comment:**

This paper studies clarification question generation for underspecified software engineering agent tasks, with the central goal of enabling agents to ask fewer but more useful and answerable questions. The authors make three main contributions: (1) a taxonomy of missing information in SWE-Bench issues and an analysis of their relative importance via Shapley value estimation, (2) an empirical study of what makes clarification questions answerable by users, and (3) a supervised + RL training pipeline that optimizes clarification generation for both task relevance and user answerability. Experimental results show that the trained model achieves performance close to fully specified task settings and matches strong GPT-5-based baselines while requiring fewer clarification questions.

Strengths:

1. Reviewers consistently praised the problem formulation as timely, practical, and highly relevant to real-world agent systems, since underspecified user requests are common in deployed software engineering environments.
2. The empirical analysis is unusually thorough: the taxonomy of missing information, Shapley-based relevance ranking, and answerability study provide valuable insights beyond the immediate method itself.
3. The proposed reward design is well motivated and methodologically thoughtful. It translates empirical findings into an RL objective that leads to strong downstream gains with an efficient 8B model.

Weaknesses and Suggested Revisions:

1. The experimental validation is still somewhat limited: the evaluation is confined primarily to SWE-Bench-derived tasks, uses a relatively small final evaluation set, and lacks broader cross-benchmark validation to establish generalization. Discussions on other agent clarification benchmarks are suggested, such as UserBench, Ask-before-Plan, etc.
2. The baseline comparisons and ablations are incomplete. Multiple reviewers noted the absence of stronger controls, especially RL variants without the proposed structured rewards, outcome-only RL baselines, and clearer SFT-only comparisons, making it harder to isolate the precise contribution of each reward component.
3. The framework relies heavily on synthetic supervision and proxy evaluation signals, including LLM-generated underspecification rewrites, GPT-based answerability judgments, and simulated users, raising concerns about whether the learned behavior will transfer robustly to genuine human-agent interactions.

After rebuttal, the concerns raised by reviewers were partially but satisfactorily addressed. In particular, reviewers acknowledged that the added ablation clarifications strengthened confidence in the reward design.

Overall, this is a solid and well-executed contribution with clear significance for interactive agent research.